# Development of Robust Varicella Zoster Virus Luciferase Reporter Viruses for In Vivo Monitoring of Virus Growth and Its Antiviral Inhibition in Culture, Skin, and Humanized Mice

**DOI:** 10.3390/v14040826

**Published:** 2022-04-15

**Authors:** Megan G. Lloyd, Michael B. Yee, Joseph S. Flot, Dongmei Liu, Brittany W. Geiler, Paul R. Kinchington, Jennifer F. Moffat

**Affiliations:** 1Department of Microbiology and Immunology, SUNY Upstate Medical University, Syracuse, NY 13210, USA; gribblem@upstate.edu (M.G.L.); liud@upstate.edu (D.L.); geilerb@upstate.edu (B.W.G.); 2Department of Ophthalmology, School of Medicine, University of Pittsburgh, Pittsburgh, PA 15213, USA; myee@krystalbio.com (M.B.Y.); jsf@pitt.edu (J.S.F.)

**Keywords:** varicella-zoster virus, antiviral drugs, bacterial artificial chromosome, luciferase, bioluminescence imaging, skin organ culture, humanized mice

## Abstract

There is a continued need to understand varicella-zoster virus (VZV) pathogenesis and to develop more effective antivirals, as it causes chickenpox and zoster. As a human-restricted alphaherpesvirus, the use of human skin in culture and mice is critical in order to reveal the important VZV genes that are required for pathogenesis but that are not necessarily observed in the cell culture. We previously used VZV-expressing firefly luciferase (fLuc), under the control of the constitutively active SV40 promoter (VZV-BAC-Luc), to measure the VZV spread in the same sample. However, the fLuc expression was independent of viral gene expression and viral DNA replication programs. Here, we developed robust reporter VZV viruses by using bacterial artificial chromosome (BAC) technology, expressing luciferase from VZV-specific promoters. We also identified two spurious mutations in VZV-BAC that were corrected for maximum pathogenesis. VZV with fLuc driven by ORF57 showed superior growth in cells, human skin explants, and skin xenografts in mice. The ORF57-driven luciferase activity had a short half-life in the presence of foscarnet. This background was then used to investigate the roles for ORF36 (thymidine kinase (TK)) and ORF13 (thymidylate synthase (TS)) in skin. The studies reveal that VZV-∆TS had increased sensitivity to brivudine and was highly impaired for skin replication. This is the first report of a phenotype that is associated with the loss of TS.

## 1. Introduction

Varicella-zoster virus (VZV) is the human neurotrophic alphaherpesvirus that causes varicella (chickenpox) during primary infection, and herpes zoster (shingles) upon the reactivation from latency in the sensory nerve ganglia. Zoster is particularly morbid, as it is frequently complicated by acute and chronic pain. While vaccines for both varicella (targeted to children) and zoster (targeted to adults 50+ years old) have decreased the incidence of the disease in the countries where they are used [1,2], VZV disease prevalence remains high worldwide. The vaccines are far from global use, and zoster vaccines still have low uptake in the countries where it can be obtained [3,4,5]. Several antiviral drugs are available to treat VZV infections, including acyclovir, foscarnet, and brivudine [6,7,8]. Much remains to be learned about VZV pathogenesis, and there is still a need for more effective and improved antiviral therapies.

The human-restricted nature of VZV greatly complicates its study, particularly as there are no effective animal models for the latency, reactivation, or reactivated disease [9,10]. Furthermore, the genetic manipulation of VZV has proven problematic, as the classical approaches that have been used for the related neurotropic virus, herpes simplex virus type 1 (HSV-1), do not work for VZV without strong selection [11]. The VZV infectivity in culture is highly cell-associated and yields low titers of cell-free virus, which are often composed of many infectious particles per plaque forming unit. This makes the purification of rare recombinants from a wild-type background difficult without a strong selection process [12,13]. The development of isogenic cloned DNA systems has provided the means to develop recombinant viruses more easily, starting with the overlapping cosmids for the vaccine virus strain (vOka) [12], and then for the parent strain of Oka (pOka) (wild type; [14,15]). Cosmid systems use 4–6 cloned DNAs that contain the complete genome as large overlapping fragments [14,15]. VZV genes are manipulated by DNA fragment subcloning, plasmid-based mutagenesis, and the rebuilding back into the cosmid. The co-transfection into permissive cells with other overlapping cosmid DNAs covering the entire VZV genome results in recombination to yield full genomes and virus, which eliminates the need for plaque purification [12].

The cosmid systems were improved with the development of VZV bacterial artificial chromosomes (BACs), where the entire VZV genome is present as a single DNA replicon [16,17]. The replication in *E. coli* required the insertion of a ~10 Kbp replicon into the genome. Two pOka BACs were developed with the insertion of the replicon into the same pOka cosmid set, which was followed by virus derivation and the retransformation of *E. coli* with circular DNA from virus-infected cells [18,19]. The VZV pOka BAC that was developed by Tischer et al. contained the BAC replicon in an internal repeat region of the genome (BAC-T1) [18]. BAC-T1 was used to generate several viruses, including VZV where VZV ORF23 and ORF62 were fluorescently tagged [20]. BAC-T1 was further refined by adding repeat sequences that flank the replicon elements to allow the self-excision of the F1 replicon (BAC DX) [18,21]. A second VZV BAC was developed by Zhu and colleagues, where the replicon was placed in the large intergenic noncoding region between ORF60 and 61 of the pOka cosmids [19,22]. However, recent studies suggest that this VZV region contains a promoter-driving mRNA that is expressed during lytic infection and viral latency [23], and it is not clear how the insertion affects the VZV replication. BAC systems have permitted the development of many recombinant viruses, which have helped inform researchers which genes are required for growth in culture, tissue, and the human host. For instance, specific gene deletions across the VZV genome suggest that the 17 to 20 ORFs are not required for growth in cell culture [24]. Many genes that are not required for growth in culture have been found to be important for growth in specific cell types or in organized tissues, such as skin and T cells [24].

The goal here was to refine the use of VZV reporter viruses for use in cultured cells, tissue explants, and humanized mouse models. Tissue explant and humanized mouse models are often limited in supply or are prohibitively costly. Thus, the use of bioluminescence to track the virus spread helps overcome the need for large numbers of mice. Firefly luciferase (fLuc) can be measured in the same sample or mouse at multiple times by using bioluminescence imaging [25]. Previously, we used a VZV that is termed VZV-BAC-Luc, which expresses the firefly luciferase from the constitutively active SV40 promoter [19,26]. Unfortunately, the luciferase expression from VZV-BAC-Luc does not necessarily reflect the VZV replication rates in tissues, as it is expressed independently of viral gene expression. Thus, we sought to identify a kinetically late promoter to drive the expression of luciferase that more accurately reflects viral replication. During this work, we also identified and corrected two unexpected mutations in the BAC-T1 and BAC-DX systems that map to ORF40, which encodes the major capsid protein; and to ORF50, which encodes glycoprotein gM. We then used a corrected BAC containing luciferase that was driven by the ORF57 gene to construct VZV lacking thymidine kinase (TK) and thymidylate synthase (TS). An analysis of the VZV ∆TK and ∆TS reporter viruses reveals insights that are important for antiviral drug development, and an unexpected phenotype for VZV TS in human skin organ culture.

## 2. Materials and Methods

### 2.1. Cells and Viruses

Human foreskin fibroblasts (HFFs) (CCD-1137Sk; American Type Culture Collection (ATCC), Manassas, VA, USA) were used up to passage 20. The human retinal pigment epithelial cell line, ARPE-19 (CRL-2302; ATCC), was grown to passage 36–40. Human malignant melanoma cells (MeWos; HTB-65; ATCC) were grown to passage 40–50. Human Tert-immortalized retinal epithelial cells (htRPE) (CRL4000; ATCC) were passaged likewise to 40. All cells were grown in Dulbecco’s Modified Eagle Medium (DMEM) with 4.5 g/L glucose, L-glutamine, and sodium pyruvate (DMEM 1X, Corning, Manassas, VA, USA), supplemented with up to 10% heat-inactivated fetal bovine serum (Benchmark FBS; Gemini Bio Products, West Sacramento, CA, USA), penicillin–streptomycin (5000 IU/mL), and amphotericin B (250 μg/mL). For the MeWo cell cultivation, the medium was supplemented with nonessential amino acids (Mediatech, Herndon, VA, USA). The viruses used in this study include: VZV pOka (parental Oka) (accession number: AB097933) [14]; VZV-BAC-Luc (from H. Zhu [19]) and VZV-ORF57-Luc [27]; VZV-ORF9-Luc; VZV-ORF14-Luc; VZV-ORF17-Luc; VZV-ORF57-∆TK; VZV-ORF57-∆TS; and VZV-ORF57-∆TK∆TS. The construct development and the viruses are shown in Figure 1. All recombinant VZV, except for VZV-BAC-Luc, were developed from BAC constructs by using the pOka BACs detailed previously [18,21].

#### 2.1.1. Generation of Luciferase Reporter Viruses

VZV luciferase reporter viruses were generated by placing the coding region of the luciferase downstream of the selected ORFs as the in-frame C protein fusions separated by a sequence encoding the T2A-directed “ribosome skipping” motif. For this purpose, a plasmid construct was developed in which the ORF for the firefly luciferase was amplified by PCR from the plasmid pGL3basic (Promega Corp) by using the primers, fLucF and fLucR (all primer sequences are presented in Table 1). The resulting DNA fragment was digested with EcoRI and BamHI (sites underlined) and was cloned into the vector, pmCherryC1 (Clontech Inc., San Jose, CA, USA), and was cut with EcoRI and BglII in order to place it in-frame with pmCherry. The resulting plasmid, pmCherryT2ALuc, contained mCherry fused in-frame to luciferase, separated by the 22 amino acid T2A motif, and it expressed functional luciferase activity from the luciferase with a single proline residue on the amino terminal end in plasmid-transfected HEK293T cells (data not shown). This plasmid was modified by cutting with EcoRI and BglII (sites downstream of the mCherry-T2A-luciferase ORF) to insert a bacterial promoter-driven zeomycin resistance gene cassette, which was amplified by PCR with primers to add BglII and terminal EcoRI flanking sites (primers: *ZeoF* and *ZeoR*; Table 1). The resulting plasmid (pmCherry-T2A-Luc-Zeo) was used as a template for the PCR amplification of the entire T2A-Luc-Zeo cassette, using primers with long 5′ extensions that added 38–40 bp flanking homology arms to the PCR product to direct the recombination into the VZV BAC-mediated l-red site-specific recombination. The methods of recombineering have been detailed by us and others previously [28,29].

The in-frame fusion to gC (encoded by ORF14), with the alteration of its stop codon to a coding residue, used the primers, gClucF2 and gClucR3. The PCR product was amplified, and the gel was purified and transformed for recombineering into pOka-DX BAC after 20 min of 42 °C heat induction by using the pGS1783 bacterial host (the kind gift of Gregory Smith, Northwestern University, Evanston, IL, USA), as detailed previously [29]. Bacterial colonies resistant to chloramphenicol and zeomycin were then validated for the BAC DNA integrity and for the correct in-frame insertion into ORF14 by restriction fragment length polymorphism (RFLP) analyses and Sanger sequencing across the fusion insert join. Virus was derived by the co-transfection of the BAC-purified DNA in MeWo or hTRPE cells, as recently described [28,29]. A similar strategy was used to generate VZV containing the T2A-luciferase-zeo^r^ that is: fused to the C terminus of ORF9, using the primers, ORF9-lucF2 and ORF9lucR3; fused to the C terminus of ORF17, using the primers, ORF17lucF2 and ORF17lucR2; and fused to the C terminus of ORF57, using the primers, 57lucF and 57lucR.

#### 2.1.2. Correction of Spurious Mutations

ORF14, 9 and 17- luciferase reporter viruses were derived from the pOkaDX BAC background and were detailed previously [18,21]. The ORF57 luciferase reporter virus was derived from a corrected version of this BAC (named, pOka DX-RR), which had been corrected for two newly identified spurious mutations that were revealed by pOka BAC DX sequencing (Table 2). One mutation in pOka DX was in ORF40, which encodes the major capsid protein and causes A428T nonsynonymous change. The ORF40 SNP was also present in the nonexcisable version of the BAC [18] but was not found in the pOka cosmids that were used to derive the Tisher BAC [14], nor in the independently derived Zhu BAC that was used to generate VZV-BAC-Luc [19]. It was not found in any other VZV sequence (of 93 genome sequences analyzed).

A second nonsynonymous mutation in ORF50, which encodes glycoprotein gM, caused V26A. The ORF50 mutation may reflect a naturally arising variant, since it was found in 4 VZV-reported sequences of the 93 scanned. The correction of the two mutations used primers that were designed to add silent restriction sites near the SNPs for the rapid identification of positive constructs, specifically a novel noncoding EcoRI site for ORF40 and a novel noncoding HindIII site for that in ORF50. The primers were used to amplify the kanamycin resistance cassette from pEPSkan2, and the gel-purified PCR products were recombined into their respective sites and were then reversed [29]. The BACs were verified for the inserted restriction sites introduced at each step and were verified by DNA sequencing across the mutations. ORF40 was corrected first, followed by the correction of ORF50, using the primers, 40REPF + 40REPR, and 50REPF + 50REPR, respectively.

#### 2.1.3. Deletion of ORF13 and ORF36 in the BAC DXRR 57luczeo Background

The partial deletion of the VZV thymidine kinase (ORF36), and the full deletion of thymidylate synthase (ORF13), used a strategy in which the genes were replaced with the mturq2blue (mT2B) fluorescent reporter gene. We first developed the plasmid, p-mTurq2Blue-KAN-in, which contained a recombination-reversable kanamycin resistance cassette derived from the plasmid, pEPS-kan2 [18]. The entire Turq2Blue-KANin cassette was then PCR amplified with the primers, TS-T2B-For and TSendR2 (to delete TS), and the primers, TKT2BFor and TKendR3 (to delete part of TK). The gel-purified PCR fragments were independently recombined into pOka BAC DX-RR57luczeo and were reversed to remove the kan^r^ cassette and to restore the mT2B ORF. This strategy placed the mT2B reporter under the control of the TS and TK promoters in each virus, respectively, when the VZV was derived as detailed above. As expected, the plaques were blue-fluorescent when illuminated with 434 nm. The partial ORF36 replacement recombined the mT2B gene in-frame with the 43rd codon of the TK ORF because the start of TK is very close to ORF35 and was suspected to contain regulatory sequences that regulate the ORF 35 expression. The partial gene replacement strategy preserved 220 bp of the sequence upstream of ORF35. To delete both the TK and TS, a kanamycin resistance cassette from pEPSkan2 was PCR amplified with the primers, TKkan-delF and TKkan-delR, and was then recombined into the BAC containing the replacement of TS with mT2B and selected for the gain of kanamycin resistance. The viruses from the BACs were derived from at least two independently derived constructs and they all showed the same phenotype in subsequent studies.

### 2.2. Compounds

The compounds used include commercially purchased cidofovir (CDV) (BEI Resources, Manassas, VA, USA), foscarnet (Millipore Sigma, Burlington, MA, USA), acyclovir (Millipore Sigma, Burlington, MA, USA), and brivudine (BVdU) (Fisher Scientific, Hampton, NH, USA). For the cell-based assays, dry compounds were dissolved in water or dimethyl sulfoxide (DMSO) and were stored at −20 °C. Stocks were diluted in complete tissue culture medium to working concentrations (CDV = 0.078–5 µM; BVdU = 0.000027–0.02 µM; ACV = 0.0026–40 µM; foscarnet = 1 mM). For the mouse experiments, CDV was prepared in water (2.5 mg/mL) and stored at 4 °C.

### 2.3. Virus Growth Kinetics in Cells

The virus growth kinetics were evaluated in HFFs or ARPE-19 cells, as previously described [26,30]. Briefly, cells were grown to confluence and were then infected with cell-free VZV at a ratio of 1:50 or 1:100, and were incubated at 37 °C. At 2 h post-infection, additional media were added to each well. The virus spread was measured with daily bioluminescence imaging. Prior to scanning, the media were removed and replaced with D-luciferin (300 µg/mL in PBS; XenoLight™ D-Luciferin Potassium Salt, Perkin Elmer, Waltham, MA, USA) for 40 min. The medium was replaced daily after imaging. The bioluminescent signal has been significantly correlated to the viral spread (pfu) in previous studies [19] and is confirmed here. VZV was added to HFF monolayers and imaged daily for bioluminescence. Cells were then harvested after imaging to determine the virus yield, measured by standard infectious center assays [19].

### 2.4. Total Flux Half-Life in Cells Infected with VZV-ORFx-Luc Viruses

Each VZV-ORFx-Luc virus and the parental VZV-BAC-Luc virus were evaluated for decay in the luciferase activity, measured as the Total Flux, in the presence of 1 mM of viral DNA replication inhibitor, foscarnet. Trypsinized and counted HFFs were resuspended in tissue culture medium and mixed with VZV-infected HFFs at a 1:50 ratio. Approximately 5 × 10^5^ HFFs with VZV in 2 mL of medium were added to 42 individual 35 mm tissue culture dishes (Corning, Corning, NY, USA) and were incubated at 37 °C in a humidified CO_2_ incubator for 36–40 h. Three dishes were scanned to measure the Total Flux at time zero (0 h). Then, 1 mM of foscarnet was added to the media in all dishes to inhibit the viral DNA polymerase and the dishes were returned to the incubator. At 2–12 h intervals, three dishes were selected at random and individually scanned by IVIS over 12–14 timepoints, which varied for each VZV reporter virus, with many clustered around the expected time when the Total Flux signals neared 50%. At each timepoint, the media were removed and replaced with D-luciferin (300 µg/mL in PBS) for 40 min at 37 °C, and were then scanned in the IVIS™ 50 instrument to record the bioluminescence, measured as the Total Flux. The triplicate values at each timepoint were divided by the average Total Flux at 0 h to calculate the percent change in the Total Flux. The time when the Total Flux reached 50% was determined by using nonlinear regression analysis to fit a dose vs. response curve (F constrained to 50).

### 2.5. Efficacy in Cultured Cells

Antiviral activity against VZV-ORF57-Luc, VZV-ORF57-∆TK, and VZV-ORF57-∆TS was evaluated by previously established dose response assays [31,32]. Briefly, ARPE-19 cells were seeded in 24-well tissue-culture-treated plates for 3 d prior to infection. Cells were infected with up to 500 pfu per well and were incubated at 37 °C with 5% CO_2_. Cidofovir and brivudine were added at varying concentrations 2 h post infection and were returned to 37 °C. Infected and treated cells were incubated for 3 d post infection. Prior to scanning, the medium was replaced with D-luciferin (300 µg/mL in PBS) for 40 min at 37 °C. The bioluminescence was measured using the IVIS™ 50 instrument (Caliper Life Sciences/Xenogen, Hopkinton, MA, USA). The virus yield, reported as the fold change, was calculated as the Total Flux (photons/sec/cm^2^/steradian) at each concentration divided by the Total Flux for the untreated wells.

### 2.6. Preparation of Skin

Human skin from fetal and adult sources was used. Fetal skin (18–20 weeks gestational age) (Advanced Bioscience Resources, Alameda, CA, USA) was obtained according to all local, state, and federal guidelines. Adult human skin was obtained with informed consent from healthy adults who were undergoing reduction mammoplasty surgeries at SUNY Upstate Medical University in Syracuse, New York. The adult human skin collection was managed under approved Institutional Review Board protocols and procedures (SUNY Upstate, Institutional Review Board #1140572), as well as all local, state, and federal guidelines. Fetal and adult skin preparation has been described previously [27,33]. Briefly, skin was cleaned with povidone iodine and ethanol and washed in sterile media. Adult skin was thinned using a Weck knife and Goulian guard (0.028”) to remove any underlying adipose and excess dermal tissue. Prepared skin was then cut into approximately 1 cm^2^ pieces and cultured on NetWells for the skin organ culture (Corning, NY, USA), or was implanted into mice, as previously described [26,34,35].

### 2.7. Skin Organ Culture

Fetal or adult human skin was cultured on NetWells (Corning, NY, USA), which hold tissue at the air–liquid interface. The skin was infected with VZV by scarification and was cultured as previously described [26,27,33]. The infected skin was incubated at 37 °C for fetal tissue, and at 35 °C for adult tissue. Virus spread was measured by bioluminescence imaging every day, or every other day, by scanning the skin in the IVIS™ 50. Prior to scanning, skin tissue was submerged in D-luciferin (300 µg/mL in PBS) for 40 min. The medium was replaced every other day after imaging.

### 2.8. Animal Procedures

Animal procedures were performed as previously described [26,35], with approved protocols, and were monitored by the Institutional Animal Care and Use Committee (IACUC) at SUNY Upstate Medical University. All studies were performed in accordance with all state and federal laws and regulations. Briefly, fetal human skin xenografts were introduced subcutaneously above the left flank of 5–6-week-old male *scid-beige* (CB.17; CB17/Icr-*Prkdc^scid^/IcrIcoCrl*) mice (Charles River, Wilmington, MA, USA). Three to four weeks postimplantation, xenografts were exposed through an incision on the mouse back and were inoculated by intradermal injection with cell-associated VZV (1 × 10^4^–10^5^ pfu/mL; 60 µL injection; grown in HFFs). Prior to imaging, mice were injected with D-luciferin (15 mg/mL in PBS) for 10 min. Virus spread was monitored by daily bioluminescence imaging with the IVIS™ 200 for 9 d. Mice were monitored daily for weight loss and signs of distress.

### 2.9. Bioluminescence Imaging

The imaging procedure was previously described in [26]. Briefly, for skin organ culture, tissue was scanned with the IVIS™ 50 instrument and images acquired for 30 s–1 min (Caliper Life Sciences/Xenogen, Hopkinton, MA, USA). For mouse studies, animals were scanned with the IVIS™ 200 instrument and images acquired for 5 min, maximum. The VZV infection was measured as the Total Flux (photons/sec/cm^2^/steradian) in a region-of-interest (ROI) encircling each skin piece or drawn over the xenograft(s). The fold change was calculated as the daily Total Flux divided by the lowest and/or initial Total Flux value.

### 2.10. Statistical Analysis

All calculations and graphs were created using GraphPad Prism (Graph-Pad Software, San Diego, CA, USA). The dose response curves were analyzed using nonlinear regression analysis. All other data was analyzed using one-way ANOVA and various post hoc tests or a Student’s *t*-test. A *p* ≤ 0.05 was considered statistically significant.

## 3. Results

### 3.1. Construction of Robust Reporter VZV Viruses

VZV-BAC-Luc was the “gold standard” reporter virus in our previous work assessing antivirals against VZV [26,35]. The luciferase reporter allows the virus growth to be measured over time in the same sample by using IVIS, which is useful in cases where considerable sample-to-sample variability may exist, or when the availability of skin pieces or mice is limited or prohibitively costly. Unfortunately, the luciferase gene in VZV-BAC-Luc is under the control of the constitutively active SV40 promoter [19], which was suspected to be rapidly transcribed upon the genome insertion into the nucleus so that the luciferase expression was not reliant on viral gene expression, and likely occurred before viral DNA replication. Here, we aimed to create improved reporter VZVs, in which the luciferase was expressed with late kinetics and, thus, was sensitive to the inhibition of viral DNA synthesis. The construction of the reporter viruses is described in detail in the methods (see Methods 2.1; Figure 1). Briefly, the luciferase gene was inserted in-frame with the C terminal residue of the VZV open reading frames (ORFs), 9, 14, 17, and 57, by using an in-frame T2A ribosome skipping motif between the VZV ORF and the luciferase gene. As luciferase is not active as a fusion protein, the T2A motif releases active luciferase with a single N terminal proline that does not affect the activity. The first-generation virus set (VZV-ORF9-Luc; VZV-ORF14-Luc; and VZV-ORF17-Luc) was found to contain mutations in ORF40 and ORF50, which were also found in the pOkaDX BAC [18]. The second-generation of VZV had the two spurious mutations of corrected and contained luciferase inserted after ORF57 (VZV-ORF57-Luc, previously described in [27]). VZV-ORF57-Luc was eventually selected as the optimal reporter virus on the basis of the experiments to follow. The third generation of VZV reporter viruses were developed from the VZV-ORF57-Luc BAC and with either VZV ORF36 (thymidine kinase (TK)), ORF13 (thymidylate synthase (TS)), or both genes, deleted to be used in drug discovery/activation mechanism studies. All viruses were viable and were assessed for the growth kinetics in cells, skin, and in SCIDhu mice to identify those with high levels of luciferase activity that would extinguish upon the cessation of viral DNA replication (such as from treatment with antiviral compounds).

### 3.2. Comparison of VZV-BAC-Luc to VZV-ORFx-Luc Reporter Viruses in Cells and Skin

It is well documented that mutations or changes to VZV ORFs can affect virulence in skin that is not necessarily observed in cell culture [15,34,36,37,38]. Thus, we conducted a comprehensive study of the VZV reporter viruses in cell culture and organized skin in order to identify virulence differences that might be a consequence of the insertion of *luc*. First, HFF cells were inoculated with cell-free VZV reporter viruses, at an approximate MOI of 0.01. In a second experiment, human fetal skin organ culture (SOC) explants were inoculated by scarification with a suspension of VZV-infected HFFs for each virus (cell-associated virus; 60 µL; 1 × 10^4^–10^5^ pfu/mL) and placed on NetWells at the air–media interface. The VZV spread in the HFFs and SOC was measured daily by bioluminescence imaging and was reported as the fold change of the Total Flux. The Total Flux (photons/sec/cm^2^/steradian) is a measure of the radiance that reflects the number of VZV-infected cells and the level of luciferase activity per cell. As expected, VZV-BAC-Luc spread exponentially during the first 4 d post infection in HFFs (Figure 2A), and the reporter activity increased through approximately 5–7 d post infection in the SOC (Figure 2B). VZV-ORF17-Luc showed a similar growth pattern that was equivalent to VZV-BAC-Luc in both the cell culture and the SOC, while both VZV-ORF14-Luc and VZV-ORF57-Luc reached a much higher Total Flux than VZV-BAC-Luc in the HFFs (Figure 2A; *p* < 0.0001; one-way ANOVA with Dunnett’s post hoc test). In the SOC, there was no apparent growth difference, with the exception that, while VZV-ORF9-Luc reached a significantly higher Total Flux in the HFFs (Figure 2A; *p* < 0.001; one-way ANOVA with Dunnett’s post hoc test), the activity was considerably lower in the SOC and declined over time (Figure 2B; not significant; *p* = 0.068; one-way ANOVA with Kruskal–Wallis post hoc test). This result suggests that the VZV ORF9-Luc virus may have growth impairments in organized skin tissue, but the other late promoter luciferase viruses were considerably improved in the signal reporter activity over the previously used VZV BAC-Luc.

We previously reported that VZV-BAC-Luc bioluminescence significantly correlated with VZV pfu [19]. We considered it necessary to confirm that the bioluminescence from the cells infected with each of the new VZV-ORF reporter viruses similarly reflected the virus spread. As such, the correlation between the Total Flux and the VZV infectivity (infectious foci or pfu) was analyzed in the HFFs daily for 4 days (Figure 2C). After the bioluminescence measurements were collected, the cells were harvested with trypsin, and the ability to form infectious centers was determined by an infectious center assay [19]. For all the VZV reporter viruses, the bioluminescence was highly correlated with the VZV spread (Figure 2C; nonlinear regression analysis with log–log line; R^2^ > 0.90). However, the pattern of luciferase activity varied for each virus. For VZV-ORF9-Luc, VZV-ORF14-Luc, and VZV-ORF17-Luc, the bioluminescence increased sharply with a marginal level of virus spread. However, for VZV-ORF57-Luc, the bioluminescence and virus spread increased gradually and in tandem, similar to VZV-BAC-Luc. This suggests that ORF57 had the optimal correlation to infectivity, where luciferase was driven by a VZV-specific late viral promoter.

It was then deemed necessary to determine the effects of halting the viral DNA synthesis on the bioluminescence from each reporter virus. The *luc* gene in VZV-BAC-Luc is under the control of the constitutively active SV40 promoter [19], and we have observed that bioluminescence persists for an extended period after the DNA synthesis is halted by antiviral compounds (data not shown). If the expression of a VZV late gene was dependent on viral DNA synthesis, then we would predict that its promoter activity might stop or decline when the replication was inhibited. To determine the Total Flux half-life after blocking the viral DNA synthesis, the HFFs were infected with VZV reporter viruses for 36–40 h, and then 1 mM of foscarnet (phosphonoformate) was added to inhibit the viral DNA polymerase during the exponential growth phase. The cultures were then scanned over an additional 96 h to measure the decrease in the Total Flux as an indicator of the luciferase activity decay. Intriguingly, the Total Flux half-life varied, depending on the ORF promoter that was regulating the luciferase expression (Figure 2D). In line with previous observations, VZV-BAC-Luc had the longest half-life, at 74.6 h, and the VZV-ORF14-Luc half-life was nearly as long, at 64 h. However, the Total Flux half-life for VZV-ORF17-Luc was intermediate, at 51 h, while VZV-ORF9-Luc and VZV-ORF57-Luc had the shortest half-lives, at 43.5 and 41.9 h, respectively. On the basis of these results and the growth patterns in the HFFs and the SOC, VZV-ORF57-Luc was selected as the optimal VZV reporter virus.

### 3.3. Comparison of VZV-BAC-Luc and VZV-ORF57-Luc in a SCIDhu Mouse Model

While VZV-ORF57-Luc performed as well as VZV-BAC-Luc in skin organ culture (Figure 2B), it was not known whether the viruses replicated equally in skin xenografts, nor if it would be responsive to in vivo antiviral treatment. We previously showed that VZV-ORF57-Luc can infect skin xenografts in SCIDhu mice [27], but the relative growth kinetics were not assessed. SCIDhu mice with human fetal skin xenografts were inoculated by intra-xenograft injection with equivalent levels of VZV-BAC-Luc or VZV-ORF57-Luc, and the spread was measured daily by bioluminescence imaging with the IVIS™ 200 instrument. The groups were vehicle alone or were treated with cidofovir (10 mg/kg), which is a broad-spectrum antiviral drug that inhibits VZV DNA polymerase [39]. The treatment was administered by intraperitoneal (i.p.) injection from DPI 3-9. The fold change in the VZV yield was calculated as the daily Total Flux divided by the average Total Flux on the lowest day for each individual mouse (typically, DPI 2 or 3). VZV-BAC-Luc grew efficiently in vivo, reaching its highest point on DPI 8, with a final fold change of 21.8 ± 8 (Figure 3A; mean ± SEM). The VZV-ORF57-Luc reporter activity was still increasing, which suggests that the virus was still growing at DPI 9, and that it had a significantly higher fold change, reaching 164 ± 63 (Figure 3B) (mean ± SEM; *p* = 0.0374; Student’s *t*-test). As expected, cidofovir significantly reduced the yield for both viruses (Figure 3; *p* < 0.05; Student’s *t*-test). VZV-ORF57-Luc’s reporter activity indicated that it replicated better in SCIDhu mice and that it was responsive to antiviral treatment, which suggests that it is robust and an optimal reporter VZV virus.

### 3.4. VZV Thymidylate Synthase (TS), but Not Thymidine Kinase (TK), Is Required for Virulence

We then used the VZV-ORF57-Luc background to construct a set of reporter viruses lacking the TK (from ORF36) and/or TS (from ORF13) genes, to evaluate the role of these nucleotide-modifying enzymes in the activation of novel antiviral compounds, and to understand their mechanism of action (Figure 1). VZV ORF36 encodes the thymidine kinase, which is required to phosphorylate many nucleoside analogs [40], while ORF13 encodes thymidylate synthase, which is involved in increasing the pool of thymidine in quiescent cells and may be involved in some nucleoside activation mechanisms [41]. Both enzymes play key roles in pathways that are affected by antiviral compounds. ORF13, ORF36, or both, were deleted and replaced with marker genes, which resulted in VZV-ORF57-∆TK, VZV-ORF57-∆TS, and VZV-ORF57-∆TK∆TS. After the genotypes were confirmed by RFLP and regional sequencing, the phenotypes were evaluated in cells and adult human skin organ culture. We previously showed that VZV grows similarly in fetal versus adult skin (data not shown) [27]. The ARPE-19 cell monolayers were inoculated with cell-free VZV at an approximate MOI of 0.01, whereas the skin explants were infected by scarification through cell-associated VZV that was placed on the skin surface (30 µL inoculum; 1 × 10^4^–10^5^ pfu/mL). The skin explants were then placed on NetWells at the air–media interface. As expected, VZV-ORF57-Luc grew well in ARPE-19 cells over 4 d, and in adult skin explants over 7 d (Figure 4). VZV-ORF57-∆TK had equivalent growth kinetics in cells and skin (Figure 4A,D). VZV-ORF57-∆TS was slightly impaired in ARPE-19 cells, but was significantly impaired in skin compared to VZV-ORF57-Luc (Figure 4B,D; *p* < 0.01; one-way ANOVA with Dunnett’s post hoc test). When both the VZV TK and TS were deleted, the growth in the cell culture was slightly less but did not reach significance (Figure 4C). However, similar to VZV-ORF57-∆TS, VZV-ORF57-∆TK∆TS showed minimal spread in skin organ culture (Figure 4D; *p* < 0.05; one-way ANOVA with Dunnett’s post hoc test). Together, these results indicate that, while TK was dispensable for skin growth under these conditions, the expression of TS may facilitate the growth and spread of VZV in confluent ARPE-19 cells and it is required for spread in skin. To the best of our knowledge, this is the first demonstration of a growth phenotype for a VZV TS-deletion mutant.

### 3.5. VZV TS Deletion Increases Sensitivity to Antiviral Drugs

Mutations in VZV TK have long been associated with antiviral resistance to acyclovir and brivudine (BVdU), which are antivirals that require an initial phosphorylation by VZV TK for activation [42]. Cidofovir does not require TK phosphorylation because of its phosphonate moiety. It is also known that blocking viral or cellular thymidylate synthase can improve the potency of nucleoside analogs, such as acyclovir [43]. However, it was not known how deletions in TK and TS would affect the antiviral sensitivity of the VZV-ORF57-Luc virus. Thus, the potency (50% effective concentration (EC_50_)) of acyclovir, brivudine, and cidofovir against VZV-ORF57-Luc, VZV-ORF57-∆TK, and VZV-ORF57-∆TS was evaluated in confluent ARPE-19 cells. The antiviral activity was assessed by bioluminescence imaging at 3 d post infection (DPI), which was previously shown to be highly correlated to the VZV pfu [19] (Figure 2C). The EC_50_ for each antiviral compound was calculated as the average Total Flux at each concentration divided by the average Total Flux of the untreated wells [26]. Here, VZV-ORF57-Luc was sensitive to both brivudine and cidofovir, with EC_50_ values of 3.0 nM and 1.7 µM, respectively (Figure 5A). As expected, VZV-ORF57-∆TK was resistant to brivudine but sensitive to cidofovir, with an EC_50_ value of 0.81 µM. Notably, VZV-ORF57-∆TS showed considerably enhanced sensitivity to both brivudine and cidofovir, with EC_50_ values of 1.9 nM and 0.81 µM, respectively. Similar to brivudine, VZV-ORF57-Luc and VZV-ORF57-∆TS were sensitive to acyclovir, with EC_50_ values of 7.6 µM and 0.37 µM, respectively (dose response curves not shown).

To further explore this phenomenon, we extended this study and evaluated the sensitivity of VZV-ORF57-∆TS to brivudine. The dose response curves indicate that the enhanced sensitivity was greatest around the EC_50_ value. ARPE-19 monolayers were inoculated with cell-free VZV, treated with 1.25, 2.5, or 5 nM of brivudine for 3 d, and the VZV yield was measured as the reporter activity. VZV-ORF57-Luc was increasingly sensitive to higher concentrations of brivudine, while VZV-ORF57-∆TK was resistant at all concentrations (Figure 5B). VZV-ORF57-∆TS was significantly more sensitive to brivudine than the parent virus at 1.25 and 2.5 nM but did not show significance at 5 nM (Figure 5B; *p* < 0.001; one-way ANOVA; Tukey post hoc test). These data further demonstrate that the absence of VZV TS enhances the antiviral sensitivity to brivudine. Given that the TK and TS deletion viruses are isogenic, and that VZV-ORF57-Luc has a robust phenotype in all the models tested, we are now poised to study the mechanism of action of the novel antivirals as they develop. These viruses should prove to be highly useful to the field.

## 4. Discussion

Here, we constructed a series of robust VZV reporter viruses by using a BAC recombineering system for use in cell culture, skin organ culture, and in SCIDhu mice, and we validated their robustness in each system. We identified a specific strong viral promoter (a VZV-unique ORF57 gene of unknown function), where the luciferase activity was linked to viral replication and had a relatively short half-life. This virus was then used to evaluate the contribution of the VZV TK and TS genes to the viral replication in the cell and skin organ culture, which identified an unrecognized role for the VZV TS gene in human skin. This work sets the stage for the further analyses of the contribution of viral genes to the VZV growth in various model systems, and for the development and testing of novel antivirals to VZV.

Several luciferase reporter VZVs have been detailed previously. These reporters are beneficial tools, as the expression of luciferase can be used with IVIS systems for the imaging of the same samples over time. Firefly luciferase is particularly useful when the tissue or host platform are heterogenous, limited, prohibitively costly, or difficult to generate, such as human skin cultures and in xenotransplants harbored in SCIDhu mice. Oliver et al. detailed the development of VZV that reported growth in SCIDhu mice by expressing luciferase fused to the ORF68 (gE) and IE63 genes [44]. However, VZV with luciferase fused to ORF68 rapidly lost its reporter activity, which is likely because of the C terminal fusion of the gE lay in the reiterated elements of the genome. The stable reporter fusion to ORF63 required the addition to both ORF63 and its duplicated gene, ORF70 [44]. When only added to ORF63, the reporter was lost. In addition, IE63 is strongly suspected to be an immediate early gene, so it may be expressed from the genome rapidly and without the need for viral replication [45]. We previously reported an alternative strategy to express luciferase, in which viral-promoter-luciferase cassettes were placed at an ectopic locus, and specifically at the ORF65-ORF66 intergenic region, in recombinant viruses [46]. While this approach was successful, we could not eliminate the possibility that the ectopic placement of the promoter region did not contain a sufficient promoter sequence to confer full regulation in the context of the virus. We also could not eliminate the possibility that the ectopic site may have influenced the regulation and expression kinetics. An advantage of the system that was developed and used here is that the native promoter at its natural locus is employed to drive the luciferase expression, which minimizes the concerns of the positional contributions to the activity. In addition, the separation of the luciferase from the viral protein with a T2A motif avoids the intolerance of the luciferase-to-protein fusion, and it ensures minimal interference with the viral protein. When fully active, the T2A ribosome skipping motif results in a 21–23 residue that is a remnant of the motif that is left attached to the viral protein, and a single proline attached to the downstream luciferase gene.

There are, however, a few potential disadvantages to this strategy, but we have yet to encounter them. Specifically, the BAC system may develop spurious mutations, or regions of the genome may be unstable in *E. coli*. Indeed, we identified two spurious mutations in the pOka DX BAC that were not present in the pOka cosmids that were used to derive the BAC, but that were present in the earlier nonself-excising version of the BAC [18,21]. We expect that this arose during the derivation of the BACs from circular viral DNA. While the deletion of the sequences has not been identified specifically for the VZV, it is recognized that part of the UL origin of the replication in the HSV genome is deleted in most *E. coli* strains [47,48], and that it may affect HSV pathogenesis in a mouse model [49]. Additionally, the potential for sporadic SNP mutations to arise is always a concern, since each recombineering step requires a clonal step, and such changes are unlikely to be detected by restriction analyses. Such mutations could accumulate with multiple recombination events, which is most often conducted by using “scarless” sequential recombineering methods, as detailed by Tischer et al. [18,21], and as employed here. Here, we acknowledge that the robust ORF57 luciferase virus underwent five sequential recombination steps (1 = the luciferase insertion; 2 and 3 = the correction of ORF40 and its reversal; 4 and 5 = the correction of ORF50 and its reversal), but we did not identify any spurious or unexpected mutations in the genome.

An important consideration when designing VZV reporter strains is the location of the *luc* insertion and the gene to which it is fused. We have found that the insertion of fluorescent proteins within VZV ORFs may alter the infectivity and virulence in skin (authors’ observations). This phenomenon has been described for other viruses, where the insertion of fluorescent reporter proteins interfered with the growth kinetics [50,51]. Thus, we specifically chose certain ORFs for the virus development. We found that *luc* insertion after the VZV ORFs, 14, 17, or 57, did not alter the growth kinetics in cells or human skin. By contrast, the *luc* insertion after the ORF9 reduced the VZV growth in skin, despite the T2A ribosome skipping motif that releases fLuc from the ORF9 protein. ORF9 interacts with IE62 to form complexes that are involved in tegument formation [52], and ORF9 is required for VZV replication [38]. It is possible that the *luc* insertion altered the ORF9 protein, thereby preventing it from interacting with the IE62 protein and performing its function during replication. Another possibility is that the *luc* sequences interrupted the mRNA and the noncoding RNA transcription through the ORF9 region [53]. While it is outside of the scope of this study, it would be interesting to create a library of VZV reporter strains with the T2A-*luc* (or other reporter genes) inserted after other candidate VZV ORFs. Such a library could potentially reveal phenotypes not previously described for different VZV ORFs. For this, the recent detailed transcriptional map of the VZV genome will be important to take into consideration [53].

A goal of this project was to construct a reporter virus with *luc* expression under the transcriptional control of a late viral promoter. The annotation of the VZV transcriptome and the kinetics of the expression, as predicted from HSV, prompted the selection of the ORF17; the true late genes, ORF9 and ORF14; and the leaky late gene, ORF57, as potential regulators of the *luc* expression [53]. Transcript abundance was also considered, since high levels would produce a brighter bioluminescence signal. ORF9 is one of the most abundant VZV proteins, it is packaged into the virion tegument, and it is detected in the cytoplasm as soon as 2 h post infection [54]. We [29] and others [55] have reported the successful tagging of ORF9 at its C terminal end, and, thus, we expected it to be a good candidate for driving luciferase. The VZV transcription kinetics of ORF14, which encodes glycoprotein C, is atypical and unpredictable in cultured cells and it may be very delayed, with little protein detected until 3 DPI, after 4–6 rounds of VZV replication [56]. ORF17, a putative host shut-off protein, is required for VZV replication at 37 °C, but not at 33 °C [24,57], although this functional role is contested [58]. ORF17 can also be detected as early as 6–12 h post infection in cells [58]. ORF57 is a very short ORF that is unique to VZV and has no known function, but it is detected in the Golgi at 48 h post infection [59]. In this study, infected cells or skin explants were first scanned at 24 h post infection, at which time there was a strong bioluminescent signal. This indicates that the fLuc protein was synthesized from the viral mRNAs that were driven by these ORFs within the first 24 h.

VZV-BAC-Luc has long been the gold standard reporter virus. Strong luciferase activity enables the bioluminescence imaging of the VZV spread, which has been instrumental in understanding VZV pathogenesis and in evaluating the antiviral compounds in cells, skin, and in vivo [19,26,27,44]. Unfortunately, *luc* is controlled by the SV40 promoter [19], which is normally considered to be constitutively expressed and may not respond to the antiviral effects of novel compounds in the context of the virus. Here, we showed that the bioluminescence half-life for VZV-BAC-Luc in the presence of the DNA polymerase inhibitor, foscarnet, is 74.6 h, or just over 3 days (Figure 2C). This is problematic during the standard antiviral dose response studies that are conducted over 72 h because the bioluminescent signal may only decrease by half in the treated infected cells. This could potentially mask the effects of an antiviral compound that is evaluated by using the VZV-BAC-Luc strain. In skin organ culture and SCIDhu mice with skin xenografts, bioluminescence imaging usually coincides with the initiation of antiviral treatment. If the bioluminescent signal is sustained for 3 days, it may conceal how quickly a novel antiviral prevents the virus spread. Thus, a new VZV reporter strain, with a shorter bioluminescent signal half-life, was needed. Here, VZV-ORF57-Luc appears to be a superior and optimal reporter strain. It had the shortest bioluminescence half-life of 41.9 h, or less than 2 days, it produced a strong bioluminescent signal, and it was virulent in skin and SCIDhu mice. The characterization of VZV-ORF57-Luc is a promising step forward and will improve the evaluation of novel antivirals.

Acyclovir and its derivatives are the first-line antiviral therapy for VZV infections, and they are activated by TK phosphorylation. Mutations in the TK gene are the most common cause of resistance to acyclovir and other nucleoside analogs that lack a phosphonate group [6,60,61]. Nucleoside analogs remain the focus of most antiviral drug development, and so we constructed VZV-ORF57-Luc-∆TK as a tool to study how nucleoside analogs are processed in VZV-infected cells. Here, we show that VZV-ORF57-∆TK was resistant to acyclovir and brivudine, which must be phosphorylated in the cell to their active form, but not to cidofovir, which has a phosphonate group. The deletion of TK had no discernable phenotype on this strain. This reporter virus is isogenic with VZV-ORF57-Luc, and it will be invaluable for screening nucleoside analogs for activity against clinical strains that are resistant to acyclovir.

VZV encodes several proteins for nucleic acid metabolism that may impact the mechanism and activity of antiviral nucleoside analogs. VZV ORFs 18 and 19 encode a ribonucleotide reductase that generates deoxyribonucleotides, and the deletion of these genes potentiates the activity of acyclovir [62]. VZV ORF8 encodes a dUTPase that converts dUTP to dUMP [63], which is a substrate for ORF13 thymidylate synthase. The TS enzyme converts dUMP to dTMP in quiescent cells [41], which is then phosphorylated by VZV TK and cellular kinases to increase the dTTP pool for viral DNA synthesis. Nucleoside analogs are also substrates of VZV TK and compete with deoxythymidine and dTMP for the enzyme. It has long been noted that inhibiting TS increases the activity of acyclovir by reducing the dTMP levels. In cells that are infected with HSV-1, blocking the viral and cellular TS activity with 5-fluorouracil increases the potency of acyclovir [43]. We found a similar effect in this study, in that the potency of brivudine increased against VZV-ORF57-∆TS. This enhancing effect was also observed for CDV and ACV, although they were not analyzed in depth. We also found that VZV-ORF57-∆TS was impaired for virus spread in skin. The cells in adult skin are mostly quiescent, with little to no cellular TS. In this environment, VZV TS may be important for generating dTMP and adequate dTTP. To the best of our knowledge, this is the first time a growth phenotype has been described for a VZV TS mutant. It would be interesting to test the VZV-ORF57-∆TS growth in replicating and quiescent cells, to measure the dTTP pools, and to discern the impact on the VZV replication and antiviral activity of nucleoside analogs. Such studies could fully resolve the VZV TS phenotype uncovered here.

In summary, we detail a strategy to generate VZV reporter viruses to assess the gene contributions to virulence in cells and skin, as well as to evaluate novel antiviral compounds. The reporter viruses were constructed from a recombinant pOka strain with a T2A ribosome skipping motif between the VZV ORF and the *luc* gene. VZV-ORF57-Luc appears to be an ideal reporter virus for its maintenance of the virulence in a human skin model and for the responsiveness to inhibition of DNA replication. We used this background to develop isogenic reporter viruses with deletions in ORF36 and ORF13 as additional tools for antiviral research. In cells and skin, we show that TK was dispensable, while TS was found to have an unprecedented requirement for virulence. Furthermore, the loss of TS enhanced the antiviral sensitivity to certain nucleoside analogues. Taken together, this work lays the foundation for the probing of the viral gene contributions to pathogenesis in human skin and for the evaluation of novel antiviral compounds.

## Figures and Tables

**Figure 1 viruses-14-00826-f001:**
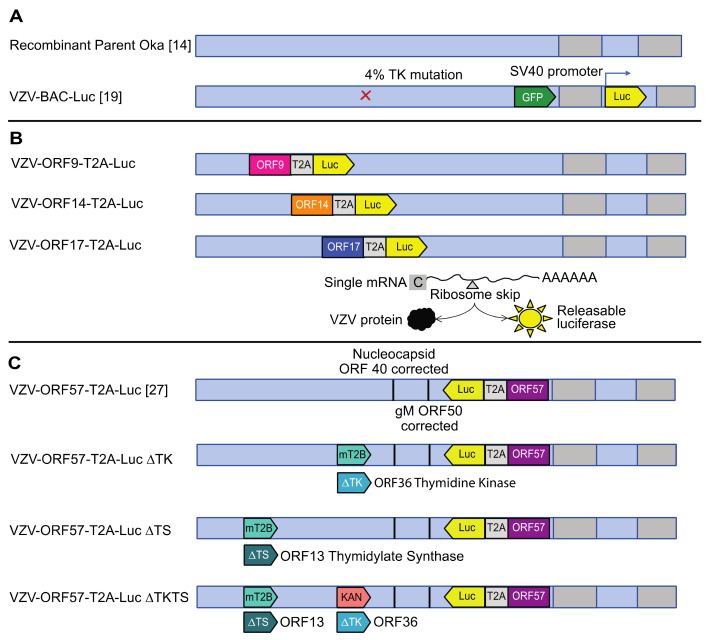
Constructs for VZV reporter viruses. The diagram depicts the presence of the recombinant sequences used to alter pOka [14] with BAC recombineering, as detailed in the text. Viruses shown in (**A**) include parent Oka and VZV-BAC-Luc, derived from the BAC-Zhu pOka system [14,19]. VZV-BAC-Luc was generated from pOka (passage 9), as previously described [19], and included a small population of TK-virus, which arose through unknown mechanisms. (**B**)—upper section: viruses derived from an uncorrected BAC system, as detailed by the Osterreider group, which include VZV-ORF9-T2A-Luc; VZV-ORF14-T2A-Luc; and VZV-ORF17-T2A-Luc. (**B**)—lower section: diagram of the viral protein and luciferase expression mediated by the T2A ribosome skipping motif, which is used to generate two proteins from one mRNA. (**C**) VZV-ORF57-T2A-Luc [27], VZV-ORF57-∆TK, VZV-ORF57-∆TS, and VZV-ORF57-∆TK∆TS were generated in a BAC with corrected spurious mutations identified in ORF40 and ORF50 (Tischer VZV-BAC [18]), as detailed in the text. In VZV-ORF57-Luc, ORF13 (TS), ORF36 (TK), or both, were deleted through site-directed recombination events to replace the partial (ORF36) or entire (ORF13) ORFs with either the mturq2blue (mT2B) fluorescent reporter gene or a kanamycin resistance cassette. The colors used for each virus are carried through the rest of the figures for continuity.

**Figure 2 viruses-14-00826-f002:**
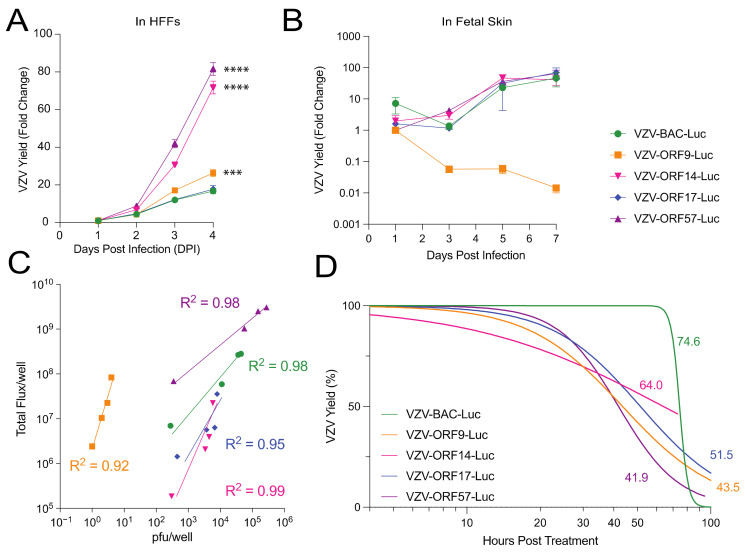
Evaluation of VZV-BAC-Luc and VZV-ORFx-Luc growth kinetics and bioluminescence in tissue culture and human fetal skin. Cells and skin were infected with cell-free or cell-associated virus, respectively, and grown at 37 °C. Cell culture experiments (**A**,**C**,**D**) were performed in HFFs, while skin organ culture (**B**) was performed in fetal skin. (**A**,**B**) VZV yield was measured by bioluminescence imaging and calculated as the fold change from the average Total Flux (photons/sec/cm^2^/steradian), divided by the lowest Total Flux value (DPI 1 for cells, or DPI 1-3 for SOC). (**C**) Correlation coefficients of luciferase and virus plaque number were determined for each VZV reporter virus on the basis of the relationship of pfu/well to average Total Flux per well. (**D**) HFFs and VZV were co-cultured for approximately 40 h prior to foscarnet treatment (1 mM) to block viral DNA replication. Values next to each curve represent the time (in h) for bioluminescence to decrease by 50% after treatment started and are shown in the corresponding color for each virus (individual points omitted for clarity of graph). Each point represents the mean ± SEM. Statistical analyses included one-way ANOVA with Dunnett’s post hoc test (**A**,**B**, *** *p* < 0.001; **** *p* < 0.0001). Nonlinear regression analysis with (**C**) log–log line or (**D**) dose response—inhibition was used for best-fit lines. *n* = 3 biological replicates for cell-based assays; *n* = 6 biological replicates for skin organ culture.

**Figure 3 viruses-14-00826-f003:**
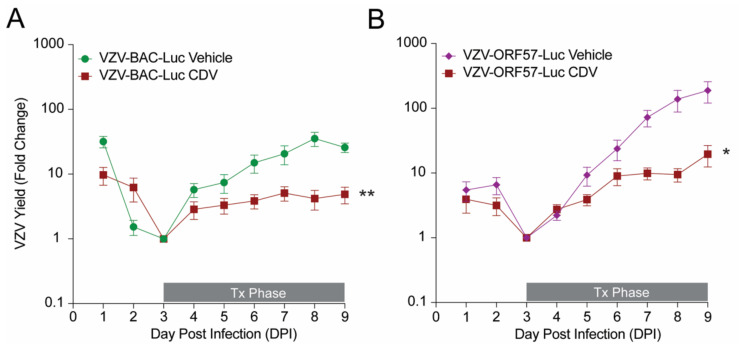
Evaluation of VZV-BAC-Luc and VZV-ORF57-Luc growth kinetics and responses to antiviral treatment in a SCIDhu mouse model. SCID *beige* mice were implanted with a single subcutaneous xenograft of fetal human skin. Xenografts were inoculated 3–4 weeks later with (**A**) VZV-BAC-Luc or (**B**) VZV-ORF57-Luc by intra-xenograft injection (1 × 10^4^–10^5^ pfu/mL; 60 µL injection; grown in HFFs). Mice were treated with vehicle (water) or 10 mg/kg/day cidofovir (CDV) by intraperitoneal injection from DPI 3 to 9 (Tx Phase). VZV yield was measured daily by bioluminescence imaging and the fold change was calculated as Total Flux each day divided by the lowest Total Flux value per mouse (usually taken on DPI 2 or 3). Virus growth kinetics were evaluated for statistical significance on DPI 9. Symbols represent mean ± SEM. Student’s *t*-test; asterisks indicate significance between vehicle and cidofovir groups (* *p* < 0.05; ** *p* < 0.01). *n* = 6–10 mice per group.

**Figure 4 viruses-14-00826-f004:**
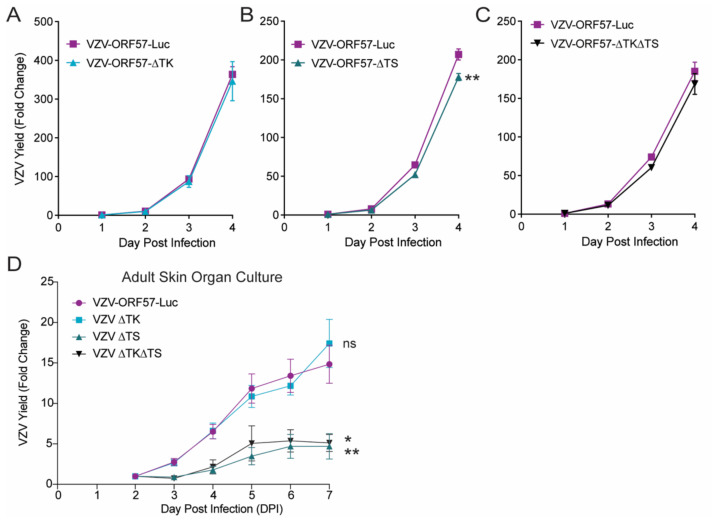
VZV TK is dispensable while VZV TS is required for optimal growth in cells and is critical for human skin organ culture. Cell-free VZV was added to ARPE-19 monolayers at 37 °C (**A**–**C**; 1:50 ratio) or introduced to adult human skin (**D**; 1 × 10^4^–10^5^ pfu/mL; 30 µL inoculum) by scarification, placed on NetWells, and incubated at 35 °C. Virus spread was monitored daily by bioluminescence imaging. Cell cultures were infected with up to 500 pfu/well. Skin was infected with 300–3000 pfu/piece of skin, depending on the viral titer that could be attained in ARPE-19 cells. VZV-ORF57-Luc was grown independently in ARPE-19 for each assay in (**A**–**C**), and directly compared to (**A**) VZV-ORF57-∆TK, (**B**) VZV-ORF57-∆TS, or (**C**) VZV-ORF57-∆TK∆TS under the same conditions. (**D**) Growth kinetics of ∆TK, ∆TS, and ∆TK∆TS in adult human skin explants were evaluated for significance compared to parental VZV-ORF57-Luc on DPI 7. VZV yield was calculated as the average Total Flux each day divided by the Total Flux on (**A**–**C**) DPI 1 or (**D**) DPI 2. Each point and line represent the mean ± SEM. (**A**–**C**) Student’s *t*-test or (**D**) one-way ANOVA with Dunnett’s post hoc test; * *p* < 0.05; ** *p* < 0.01, ns = not significant. *n* = 6 biological replicates.

**Figure 5 viruses-14-00826-f005:**
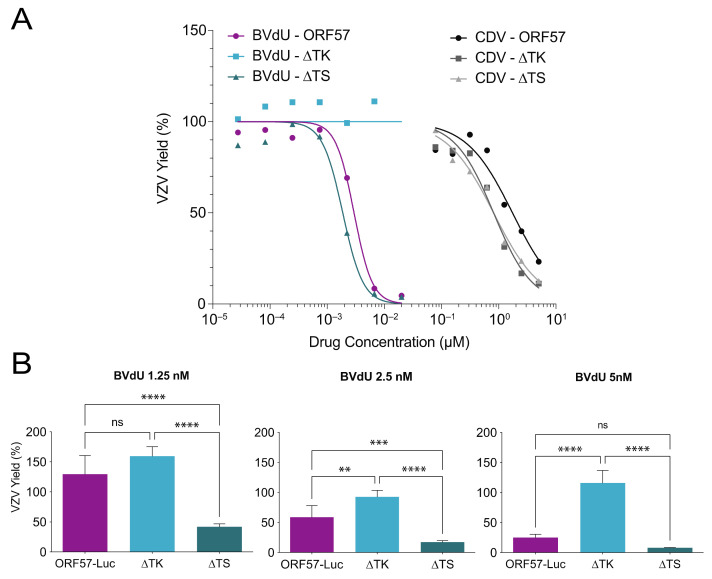
Loss of VZV TK and TS affect virus sensitivity to select antivirals in cell culture. Cell-free VZV was added to ARPE-19 monolayers at 37 °C and compounds were added at 2 h postinoculation (HPI). Antiviral compounds remained in culture medium until VZV yield was measured on DPI 3. Infected cells were treated with cidofovir (CDV) and brivudine (BVdU) (**A**) in a range of concentrations, or (**B**) at specific concentrations of BVdU (1.25, 2.5, or 5 nM). VZV yield was calculated as the average Total Flux of treated wells divided by the average Total Flux of untreated wells. (**A**) Each point represents the mean Total Flux with the best fit line. (**B**) Each bar represents the mean + SD. Asterisks indicate significance between VZV viruses for each treatment condition (ns = no significance; ** *p* < 0.01; *** *p* < 0.001; **** *p* < 0.0001). *p* < 0.05; one-way ANOVA; Tukey post hoc test. *n* = 3 (**A**) to 6 (**B**) biological replicates.

**Table 1 viruses-14-00826-t001:** Primers used to generate VZV viruses.

Gene Target	Forward Primer (5′-3′)	Reverse Primer (5′-3′)
fLuc Firefly (Photinus pyralis) Luciferase	GAGGGATCCGGTTCCGGAGAGGGCAGAGGAAGTCTGCTAACATGCGGTGACGTCGAGGAGAATCCTGGCCCAATGGAAGACGCC AAAAACATA	AATTCGAATTCGCGCGCAGATCTTTACACGGCGATCTTTCCGCCCTTCTTGGC
Zeo (Zeomycin resistance)	AGATCTAGATCT*C*GAGTAATGGAACGGACCGTGTTGAC	GCTGACGTCGACGAATTCTGATCACTCAAGTTTCGAGGTCGAGGTG
gCLuc (F2 and R3; ORF14)	CTTATCGCAGTTATCGCAACCCTATGCATCCGTTGCTGTTCAATGGACGAGCTGTACAAG	AATAAAATGATATACACAGACGCGTTTGGTTGGTTTCTGTCAGTCCTGCTCCTCAGCC
ORF9Luc (F2 and R3)	AGTAGGGCCCGTTCGGCATCAAGAACTGATGCGCGAAAATCAATGGACGAGCTGTACAAG	ACGTTTATTTATTATACATAATACCGGGTAAACCGTTACTTCAGTCCTGCTCCTCAGCC
ORF17Luc (F2 and R2)	CTCCACTCCCACTAAACACTGTATTAACAAAATATTGGAATTCAATGGACGAGCTGTACAAG	AGCAAAATAAAACAATGAACCATTAAGTCGCTCTTATGTGTGTCAGTCCTGCTCCTCAGCC
57Luc (ORF57)	ACGTTGAGGAGCCTTGCAGGTTGGGTGCCGCGCTTCACCGTCAATGGACGAGCTGTACAAG	TTTATATTTAACGGCTTTTAATTTGAAGACACCTATCCTCTCAGTCCTGCTCCTC
40REP (ORF40)	ATATAGATATTACGTTTATCATGCCAATGGGAGTGTTTCAGGCGAATTCCATGGACAGATATACACGACAGGATGACGACGATAAGTAGG	CAGTTGAAAAATCGCCGGCGTGTCGTGTATATCTGTCCATGGAATTCGCCTGAAACACTCCCATTGGCACAGGGTAATGCCAGTGTTAC
50REP (ORF50)	TGAATTATCCAAATTCGCCAATTAAGCGTATCCATTTGATGATCTAAAGCTTCCACCTCGGGTGTCGTGGTGTCGTACGGGGATGACGACGATAAGTAGG	TCTGAAAAAGTCTCACCGTACGACACCACGACACCCGAGGTGGAAGCTTTAGATCATCAAATGGATACGCTTAATAGGGTAATGCCAGTGTTAC
TS (TS T2B-For and endR2)	TATCAAGTGGTCGTTTGTATTTAACGATTATTACCGGTACCGGTACCATGGTGAGCAAGGGCGAGGAG	ACATCTACTGTCTTGACAACATTTAAAAATCCATTAAAGATTATCTGGATCCTACCTTTC
TK (TK T2BFor and endR3)	CGCCGAAGAATTTTTACACCACTTTGCAATAACACCAAACGGTACCATGGTGAGCAAGGGCGAGGAG	TGTGTATCATCTTTTTACTGGTACATACGTAAATACTAGGTTATCTGGATCCTACCTTTC
TKkan-del (For and Rev)	CGCCGAAGAATTTTTACACCACTTTGCAATAACACCAAACGGATGACGACGATAAGTAGG	TGTGTATCATCTTTTTACTGGTACATACGTAAATACTAGGAGGGTAATGCCAGTGTTAC

The restriction sites engineered to sequences to enable cloning, as detailed in the text, are underlined.

**Table 2 viruses-14-00826-t002:** Spurious mutations detected in pOka-DX BAC and their corrections.

Location of Mutation	Identification of Mutation and the Correction
ORF40 mutation in pOka DX	72661 atg cca atg gga gtg ttt cag **A**ca aac tcc atg gac aga tat aca cga cac **…** M P M G V F Q **T_428_** N S M E R Y T R H
ORF40 Correction in pOka DX-RR	72661 atg cca atg gga gtg ttt cag **G**cG aaT tcc atg gac aga tat aca cga cac … M P M G V F Q **A_428_** N S M E R Y T R H
ORF50 mutation in pOka DX	gaa aaa gtc tca ccg tac gac acc acg aca ccc gag g**C**g gaa gcg tta gat cat caa atg E_11_ K V S P Y D T T T P E **A** E A L D H Q M
ORF50 correction in pOka DX-RR	gaa aaa gtc tca ccg tac gac acc acg aca ccc gag g**T**g gaa gcT tta gat cat caa atg E_11_ K V S P Y D T T T P E **V** E A L D H Q M

Note: Lower case letters denote the sequence identical to the parent, while capital letters represent the spurious mutation in pOka DX (**bolded**), or the changed residues made to accompany the correction to introduce a novel restriction site (underlined). The single letter amino acid code of the region of the protein affected is shown below each sequence, with the key mutant residue **bolded**.

## Data Availability

The data in this study are fully presented in this article. These data are also available upon reasonable request from the corresponding authors.

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
