# Peer review of "Development of Robust Varicella Zoster Virus Luciferase Reporter Viruses for In Vivo Monitoring of Virus Growth and Its Antiviral Inhibition in Culture, Skin, and Humanized Mice"

_viruses, 2022, doi:10.3390/v14040826_

Round 1

Author Response

Response to Reviewer 1:

I found weakest part of the study to be the lack of electron microscopic analysis. The case for linkage of virus growth to light production would be strengthened by demonstrating that the density virion images in EM pictures correlated with the luciferase signal.

We are grateful the reviewer appreciates that firefly luciferase is a useful reporter gene for measuring VZV spread by bioluminescence imaging. We also thank them for their suggestion to correlate the luciferase signal to virion density measured by electron microscopy. We considered this carefully and concluded that this approach is not likely to give us meaningful data. The first consideration is that firefly luciferase is active in the cytoplasm of VZV-infected cells, and the bioluminescence intensity is proportional to the amount of enzyme. Luciferase is not incorporated into the nascent virions. The second consideration is that VZV assembly is aberrant in cultured cells, with many defective particles produced in vesicles and attached to the plasma membrane. The particle to pfu ratio has been estimated at 40,000 (Carpenter et al, 2009, PMID: 19369328). Counting virion density, either by scanning or transmission electron microscopy, would be futile due to the preponderance of defective particles. In addition, quantification of virions by EM is difficult because images offer a partial, 2-dimensional view of the infected cells, which are clustered in foci in the cell monolayer. Thus, accurate quantification depends on where the sample is taken and the field of view. For these reasons, we concluded that luciferase activity is not likely to correlate with virion number. On the other hand, we clearly show that luciferase activity is correlated with VZV pfu, which is a measure of infectivity.

Reviewer 2 Report

Lloyd et al. developed varicella-zoster virus (VZV) luciferase reporter viruses to analyze the phenotypes of VZV in cell cultures as well as a mouse xenograft system. A reporter gene, f-luc was placed in frame at the 3’-end of several candidate genes after a “ribosome skipping” motif preceding f-luc. Thus F-Luc is expressed as a free protein under the control of natural viral promoters. Specifically, they tested fusion with ORF9, 14, 17, and 57 to generate reporter viruses (VZV-ORFx-Luc). They found VZV-ORF57-Luc was suitable as a reporter virus. Using this background, they generated deletion mutants of TK and TS to show the usefulness of their reporter viruses for the analysis of drug sensitivity. Given the importance of VZV infection in the field as well as clinical settings, their new reporter will be a useful tool to investigate the pathogenesis of VZV and development of antivirals.

Authors should address the following points:

  1. In the text in line 348, authors described that VZV-ORF14-Luc showed a similar growth pattern equivalent to VZV-BAC-Luc…, while VZV-ORF17-Luc and VZV-ORF57-Luc reached much higher ….” However, in Fig2A and B, VZV-ORF17-Luc (not 14) behaved like VZV-BAC-Luc, and VZV-ORF-14 (instead of 17) and 57 reached higher bioluminescence. Authors should check the description.
  2. In the Fig 2 legend, it was said that VZV yield was expressed as fold change with the value of DPI 1 as a reference, then why the values in Fig2B on DPI1 were not 1 for all the constructs?

Author Response

Responses to Reviewer 2:

In the text in line 348, authors described that VZV-ORF14-Luc showed a similar growth pattern equivalent to VZV-BAC-Luc…, while VZV-ORF17-Luc and VZV-ORF57-Luc reached much higher ….” However, in Fig2A and B, VZV-ORF17-Luc (not 14) behaved like VZV-BAC-Luc, and VZV-ORF-14 (instead of 17) and 57 reached higher bioluminescence. Authors should check the description.

We thank the reviewer for catching this mistake in the text and agree that ORF14 and ORF17 were inadvertently flip-flopped. This has been changed in the text to accurately represent the data presented in Figure 2.

In the Fig 2 legend, it was said that VZV yield was expressed as fold change with the value of DPI 1 as a reference, then why the values in Fig2B on DPI1 were not 1 for all the constructs?

We thank the reviewer for their question about fold change values in relation to the figure caption text. We apologize for the confusion, as total flux in cells is calculated based DPI 1, resulting in a value of 1. However, for SOC (and mice), cell-associated virus is used to infect the skin. Thus, we use the lowest total flux value to calculate the fold change (unless the signal continual drops, as was the case for VZV-ORF9-Luc). The lowest total flux value is usually DPI 1-3 when luciferase from the initial inoculum has been degraded and better reflects the infected cells in the skin tissue. This is why some of the fold change values are higher on DPI 1 and then reach a value of “1” on DPI 3. We have changed the figure caption to reflect this (line 375-377): “VZV yield was measured by bioluminescence imaging and calculated as fold change from the average Total Flux (photons/sec/cm2/steradian) divided by the lowest Total Flux value (DPI 1 for cells or DPI 1-3 for SOC).”